# Microglia-like Cells Promote Neuronal Functions in Cerebral Organoids

**DOI:** 10.3390/cells11010124

**Published:** 2021-12-30

**Authors:** Ilkka Fagerlund, Antonios Dougalis, Anastasia Shakirzyanova, Mireia Gómez-Budia, Anssi Pelkonen, Henna Konttinen, Sohvi Ohtonen, Mohammad Feroze Fazaludeen, Marja Koskuvi, Johanna Kuusisto, Damián Hernández, Alice Pebay, Jari Koistinaho, Tuomas Rauramaa, Šárka Lehtonen, Paula Korhonen, Tarja Malm

**Affiliations:** 1A.I. Virtanen Institute for Molecular Sciences, University of Eastern Finland, 70211 Kuopio, Finland; ilkka.fagerlund@uef.fi (I.F.); antonios.dougalis@uef.fi (A.D.); anastasia.shakirzyanova@uef.fi (A.S.); mireia.gomez.budia@uef.fi (M.G.-B.); anssi.pelkonen@uef.fi (A.P.); henna.konttinen@uef.fi (H.K.); sohvi.ohtonen@uef.fi (S.O.); feroze.fazaludeen@uef.fi (M.F.F.); marja.koskuvi@helsinki.fi (M.K.); jari.koistinaho@helsinki.fi (J.K.); sarka.lehtonen@uef.fi (Š.L.); paula.korhonen@uef.fi (P.K.); 2Neuroscience Center, University of Helsinki, 00014 Helsinki, Finland; 3Institute of Clinical Medicine, Internal Medicine, University of Eastern Finland and Kuopio University Hospital, 70211 Kuopio, Finland; johanna.kuusisto@uef.fi; 4Department of Anatomy & Neuroscience, University of Melbourne, Melbourne, VIC 3010, Australia; Damian.Hernandez@unimelb.edu.au (D.H.); apebay@unimelb.edu.au (A.P.); 5Department of Surgery, University of Melbourne, Melbourne, VIC 3010, Australia; 6Centre for Eye Research Australia, Royal Victoria Eye and Ear Hospital, Melbourne, VIC 3002, Australia; 7Department of Pathology, Kuopio University Hospital, 70029 Kuopio, Finland; tuomas.rauramaa@uef.fi; 8Unit of Pathology, Institute of Clinical Medicine, University of Eastern Finland, 70210 Kuopio, Finland

**Keywords:** cerebral organoid, microglia, neurogenesis, neuronal function, development

## Abstract

Human cerebral organoids, derived from induced pluripotent stem cells, offer a unique in vitro research window to the development of the cerebral cortex. However, a key player in the developing brain, the microglia, do not natively emerge in cerebral organoids. Here we show that erythromyeloid progenitors (EMPs), differentiated from induced pluripotent stem cells, migrate to cerebral organoids, and mature into microglia-like cells and interact with synaptic material. Patch-clamp electrophysiological recordings show that the microglia-like population supported the emergence of more mature and diversified neuronal phenotypes displaying repetitive firing of action potentials, low-threshold spikes and synaptic activity, while multielectrode array recordings revealed spontaneous bursting activity and increased power of gamma-band oscillations upon pharmacological challenge with NMDA. To conclude, microglia-like cells within the organoids promote neuronal and network maturation and recapitulate some aspects of microglia-neuron co-development in vivo, indicating that cerebral organoids could be a useful biorealistic human in vitro platform for studying microglia-neuron interactions.

## 1. Introduction

Three-dimensional cell cultures have emerged as a novel platform to study cell functions in a microenvironment resembling native tissue, with particular focus on producing transplants for regenerative medicine and screening drug compounds in biorealistic human in vitro tissue. The advances in bioprinting multiple cell types on biomaterial scaffolds [1], and generation of organoids, derived from embryonic or induced pluripotent stem cells (iPCSs) and recapitulating the overall structure and function of the organ of interest, have delivered unprecedented opportunities to study the development and physiology of complex organs such as the gut, the skin and the brain [2,3,4,5,6,7,8]. Furthermore, drug development programs may employ the technology for drug screening and precision medicine [9,10,11,12,13,14] and functional assays in vitro, such as migration of cancer cells [15,16] and passage of drug candidates through the blood-brain-barrier [17,18].

Human brain organoids recapitulate the tissue architecture of the developing cortex to a remarkable degree of fidelity, even discriminating species differences in the development dynamics of neurons between humans and gorilla [19]. Since the initial nearly parallel work by three research groups [20,21,22], brain organoids have served in modelling various brain diseases [23,24,25,26], including microcephaly [20], glioblastoma [27,28,29], Zika-virus infection [30] and Alzheimer’s disease [31,32,33].

The accumulating body of scientific reports portray brain organoids as a human 3D in vitro platform that achieves remarkable complexity, even resembling its in vivo counterpart, the embryonic human brain [34,35,36,37,38]. The main cell types of the brain, astrocytes and neurons, spontaneously develop through ectodermal lineage in brain organoids and self-assemble to form cortical layers resembling human brain organization [39]. However, microglia, the brain’s immune cells, do not innately populate brain organoids, since they originate from mesodermal lineage, unlike other brain cells [40,41,42]. In addition to their roles in immunity, microglia participate in forming and moulding the CNS at different stages of development from neurogenesis to maturation of synapses and neural networks [43,44,45,46]. During early development, microglia restrict the number of neuron progenitors arising at the subventricular zone (SVZ) [43] and support the differentiation of neurons, astrocytes and oligodendrocytes [47,48]. The dual role of microglia is evident also later in development. Microglia support the formation of synapses through direct contact [45] and indirectly through secretion of cytokines [49,50]. Yet, they restrict the number of neuronal connections by phagocytosing inactive synapses through the complement cascade [44,51]. Microglia are also reported to modulate the formation of neural circuits [46,52]. In addition to neurodevelopment, microglia have a distinguished role in the progression of neurodegenerative diseases [53]. Thus, the past years have seen multiple approaches to incorporate microglia into brain organoids.

The first such report showed microglia, derived from induced pluripotent stem cells, to migrate spontaneously from culture medium to the brain organoid. Therein, they assumed their native functionality by migrating to the site of tissue injury, induced by a needle prick [54]. Recently, the inflammatory functions of incorporated mature embryonic stem cell derived microglia were also assessed in tubular cortical organoids [55]. In an alternative approach, subtle adjustment to one of the original organoid protocols [20] gave rise to a native population of microglia capable of immune response and phagocytosis of synaptic material, as shown by internalisation of post-synaptic density protein 95 (PSD95) [56]. In another report, the incorporation of mesodermal progenitors introduced vasculature into the organoid, and remarkably, also microglia-like cells [57]. Likewise, co-clustering of microglia progenitors and neural progenitor cells gave rise to cortical organoids with microglia-like population capable of immune response and synaptic pruning [58].

Earlier work in characterising the fundamental neurophysiological properties of organoids in absence of microglia have addressed the maturation of sodium/potassium currents [22,30], sustained firing [22,59,60] and synaptic activity [22,30,61,62] in patch-clamp recordings, and of spontaneous bursting [37,60,62,63,64] and oscillatory activities using MEA recordings [64]. A very recent electrophysiological report showed that transplantation of human primary microglia to brain organoids (at 15th week in vitro) gave rise to synchronous network burst activity using a single developmental time point read-out [65]. 

However, none of previous studies above focused specifically on the emerging, functional, single-cell electrophysiological properties of neuronal entities grown in organoids and the role of microglial enrichment in modulating those functional properties over developmental time in vitro. Thus, while iPCS-derived microglia in organoids have recently been increasingly well characterised, their role in emergance of neuronal functions merits further exploration. 

Herein, we carry out a detailed single-neuron interrogation of electrophysiological functions that arise in brain organoids in response to incorporation of microglia-like cells. By evaluating neuronal functions at multiple developmental time points, we show, for the first time, that organoids enriched with microglia-like populations develop neurons that respond to a depolarising stimulus with sustained repetitive firing, while neurons without enrichment are more prone to single spiking and adaptation. Remarkably, neurons co-maturing with microglia-like cells showed a greater diversity in the expression of distinct neuronal currents, including a prominent post-inhibitory low-threshold current leading to post-inhibitory spiking, an action attributed to T-type calcium channels, which have been implicated in a variety of functions, including synaptic plasticity and neuronal differentiation [66,67,68]. We also found that excitatory post-synaptic currents (EPSPs), evidence of synaptically connected neurons, were present only in neurons from organoids with microglia-like cells. Finally, targeted single-cell and multielectrode array recordings revealed that spontaneous and NMDA-induced neuronal bursting activity was stronger and more prevalent in organoids with microglia-like cells, while NMDA-induced spectral changes in local field potentials were more significantly prominent in the γ-band frequency range in organoids enriched with microglia-like cells. Taken together, these data suggest that multiple aspects of single-neuron functional characteristics are strongly impacted by the presence of microglia-like cells. Our work herein consolidates brain organoids as a tool to study the roles of microglia in brain development, in ways not accessible in vivo. 

## 2. Materials and Methods

### 2.1. Human iPSC Lines and Ethical Considerations

Four iPSC lines were used and their phenotype and origin are presented in Table 1. Three of the iPSC lines, MBE2968, Bioni010-C2 and Ctrl8 c2, were previously characterized [69,70,71,72]. The Mad6 line was generated from fibroblasts (skin biopsy and subsequent work under Research Ethics of Northern Savo Hospital district (license no. 123/2016)) that were first expanded in Iscove’s DMEM media containing 20% fetal bovine serum, 1% Penicillin-Streptomycin and 1% non-essential amino acids. For reprogramming (all Thermo Fisher Scientific, Waltham, MA, USA) the cells at 90% confluency were transduced using CytoTune™-iPS 2.0 Sendai Reprogramming Kit (Thermo Fisher Scientific). The media was changed 24 h after transduction and then daily. At day 6, fibroblast culture medium was replaced with Essential 6 Medium (E6, Thermo Fisher Scientific) supplemented with 100 ng/mL basic fibroblast growth factor (bFGF). The cells were re-plated on the next day to 6-well Matrigel-coated plates with a density of 60,000 cells/well. The individual colonies were picked to 24-well Matrigel-coated plate containing Essential 8 Medium (E8, Thermo Fisher Scientific) between days 17–28 and passaged with 0.5 mM EDTA weekly. One week later, three colonies were selected and expanded on 6-well plates with daily media changes. For RT-qPCR, (Appendix A), total RNA was extracted (RNeasy Mini kit, Qiagen, Hilden, Germany), cDNA synthesized using Maxima reverse transcriptase (Thermo Fisher Scientific) and Taqman probes (Table 2). For ICC (Appendix A), the iPCS colonies were fixed (4% paraformaldehyde, 20 min at RT, permeabilized with 0.2% Triton X-100 (in case of Nanog and 4 October), blocked in 5% normal goat serum one hour in RT and incubated with the primary antibodies (Table 3) O/N at 4 °C, followed by secondary antibody incubation one hour at RT. Images were taken at Zeiss Axio microscope (Carl Zeiss AG, Oberkochen, Germany), scale bar 100 µm (Appendix A). For embryoid body (EB) generation, iPSCs colonies were detached by a scalpel to ultra-low adherent dishes (Corning, Corning, NY, USA) and cultured in DMEM media (Thermo Fisher Scientific) supplemented with 20% serum replacement and 1% Penicillin-Streptomycin. The EBs were plated onto Matrigel-coated 24-well plates and left to differentiate for two weeks (Appendix A). Expression of markers for all three embryonic layers was performed by flow cytometry (Appendix A) (CytoFlexS, Beckman Coulter, IN, USA). The EBs were dissociated using accutase (4 min, at 37 °C followed by trituration into single cell suspension, centrifuged 300× *g* 5 min, resuspended and filtered through 35 µm strainer. Cell viability was assessed by Zombie Aqua (Biolegend, San Diego, CA, USA), cells fixed in 1% formaldehyde, permeabilised in 0.2% TritonX + 5% FBS, and incubated with fluorophore-conjugated antibodies (Table 3) 30 min at RT. Karyotyping performed at Synlab oy (Finland) (Appendix A).

### 2.2. iPSCs Culture

iPSCs were maintained in serum- and feeder-free conditions in Essential 8 Medium (E8, #A15169-01, Thermo Fisher Scientific) supplemented with 0.5% penicillin-streptomycin (P/S, #15140122 Thermo Fisher Scientific) on growth factor reduced Matrigel (Corning) at 5% CO_2_ and 37 °C. Medium was completely changed every 24 h. Twice a week, when cultures reached 80% confluency, the cells were passaged as small colonies with 0.5 mM EDTA (Thermo Fisher Scientific) in Ca^2+^/Mg^2+^-free DPBS (Thermo Fisher Scientific). The medium was supplemented with 5 µM of Y-27632 (#S1049, Selleckchem, Houston, TX, USA), a Rho-Associated Coil Kinase (ROCK) inhibitor (#S1049, Selleckchem,), for the first 24 h to prevent apoptosis after passaging. Cultures were tested for mycoplasma using a MycoAlert Kit (Lonza, Basel, Switzerland)), and for bacteria, mould, fungi and yeast using solid and liquid lysogeny broth (LB, Merck, KGaA, Darmstadt, Germany) and sabouraud dextrose agar (SDA, VWR, Radnor, PA, USA) at 30 °C and 37 °C. 

### 2.3. Organoid Differentiation

Cerebral brain organoids were differentiated according to a protocol published earlier [20], with some modifications. Embryoid body differentiation was initiated on day 0 by detaching 70–80% confluent iPSCs with 0.5 mM EDTA and seeding 9000 cells/well on ultra-low attachment U-bottom 96-well plates (CellCarrier-96 Spheroid ULA, #6055330 Perkin Elmer, Waltham, MA, USA) in 150 µL of E8 + 0.5% P/S supplemented with 20 µM ROCK inhibitor. On days 1, 3 and 5, 120 µL of medium was removed and 150 µL fresh E8 + 0.5% P/S was added. Neuroinduction was mediated by a gradual transition from E8 to neuroinduction medium by replacing 150 µL of the medium on days 6, 7, 8 and 10 to neuroinduction medium consisting of DMEM (Thermo Fisher Scientific #21331-020), 1× N2 (Thermo Fisher Scientific #17502-001), 1× Glutamax (Thermo Fisher Scientific #35050-038), 1× of non-essential amino acids (NEAA; Thermo Fisher Scientific #11140-035), 5 U/mL of Heparin (LEO Pharma, Ballerup, Denmark) and 0.5% P/S. On day 11, the spheroids were embedded to 20µL of Matrigel and cultured in differentiation medium 1 (Diff1, 1:1 mixture of DMEM F12 and Neurobasal (Thermo Fisher Scientific #21103-049), 1× Glutamax, 0.5× NEAA, 0.5× of N2, 1× of B27 without vitamin A (Thermo Fisher Scientific #12587010), 2.5 µg/mL of insulin (#10516-5ml, Merck), 50 µM of 2-mercaptoethanol (Merck) and 0.5% P/S). On d15, organoids were transferred onto 6-well plates (Sarstedt, Nümbrecht, Germany) in differentiation medium 2 (Diff2) with identical composition with Diff1,but supplemented with B27 containing vitamin A (Thermo Fisher Scientific; #17504001) instead of with B27 minus vitamin A (Thermo Fisher Scientific #12587010). From now on, organoids were maintained on orbital rotators (ThermoFisher Scientific #88881102) adjusted to 75 rpm. Medium was replenished every second day. To promote a good quality of organoids, the morphology of the spheroids was assessed under a microscope one-by-one before embedding to Matrigel. The spheroids accepted for Matrigel embedding were translucent, with some darkening of the core permitted, and maintained sharp edges. Common exclusion criteria were (i) local expansion of tissue bulging out from the spheroid, i.e., not sharp edges (ii) a foreign fibre translodged in the spheroid, (iii) incomplete formation of the embryoid body, where the well housed one main spheroid and one or more smaller spheroids attached to the bigger one. Moreover, on day 15, organoids without visible cortical loop structures were discarded. Finally, at any point on day 15+, organoids where the dense core had disintegrated, leaving a translucent piece of tissue, were also discarded.

### 2.4. Differentiation of Erythromyeloid Progenitor Cells

To initiate microglial differentiation on day 0, iPSCs were detached with 0.5 mM EDTA at 70–80% confluency, and 60,000–80,000 single cells/dish were seeded on 3.5 cm matrigel coated dish (Sarstedt) in mesodermal differentiation medium consisting of E8, 0.5% P/S, 5 ng/mL BMP4 (PeproTech, London, UK), 25 ng/mL Activin A (PeproTech), 1 μM CHIR 99021 (Axon Medchem, Groningen, The Netherlands) and 10 μM ROCK inhibitor (Selleckchem #S1049). Cultures were maintained in a low oxygen incubator at 5% O_2_, 5% CO_2_ and 37 °C. During the differentiation, medium was changed every 24 h. On day 1, the mesodermal medium was replaced but with lower 1 μM Y-27632. On day 2, within 44–46 h from seeding, hemogenic endothelial differentiation was induced by replacing medium with Dif-base medium consisting of DMEM/F-12, 0.5% P/S, 1% GlutaMAX, 0.0543% sodium bicarbonate (all from Thermo Fisher Scientific), 64 mg/L L-ascorbic acid and 14 μg/L sodium selenite (both from Sigma), and supplemented with 100 ng/mL FGF2, 50 ng/mL VEGF (both from PeproTech), 10 μM SB431542 (Selleckchem), and 5 μg/mL insulin (Merck). On day 4, erythromyeloid differentiation was furthered by replacing the media by Dif-base supplemented with 5 μg/mL insulin, 50 ng/mL FGF2, vascular endothelial growth factor (VEGF), interleukin 6 (IL-6) and thrombopoietin (TPO), 10 ng/mL IL-3 and stem cell factor (SCF; all Peprotech). From now on, cells were maintained at atmospheric 19% O_2_, 5% CO_2_ and 37 °C. On day 8, the erythromyeloid progenitors (EMPs) that were blooming into suspension were collected by gentle pipetting through a 100 µm strainer (VWR). The cells were counted and centrifuged at 160× *g* for 5 min to be ready for incorporation into organoids.

### 2.5. Incorporation of Microglial Progenitor Cells into Organoids

To incorporate exogenous microglia into the organoids, day 8 microglial progenitors (EMPs) were allowed to spontaneously migrate into day 30 organoids using two comparable methods: either as free-floating cells or inside Matrigel droplet. For free-floating incorporation, single organoids were transferred to ultra-low attachment 96-well U-bottom wells (CellCarrier-96 Spheroid ULA, Perkin Elmer 6,055,330) in carryover medium. Then 500,000 EMPs/organoid, in 100 µL of Diff2, were seeded on the wells. For control, Diff2 was added without progenitors. Organoids with and without microglia, denoted (+)ORG and (−)ORG, respectively, were incubated at 37 °C, 5% CO_2_ for 6 h to allow the progenitors to sediment and attach to the surface of the organoid. Then organoids were transferred back to 6-well format and the culture continued according to organoid maintenance protocol. For Matrigel-droplet incorporation method, day 30 organoids were transferred to parafilm cups and day 8 microglial progenitors (EMPs) were resuspended in Matrigel at 10,000 cells/µL density. Then 2 µL of the suspension was added on an organoid, giving 20,000 EMPs/organoid. The assemblies were then incubated at 37 °C, 5% CO_2_ to allow the Matrigel to solidify and settle on the organoids. The organoids were transferred to Diff2 the same way as on day 11 and the culture was then continued according to organoid maintenance protocol.

### 2.6. Organoid Fixation and Immunohistochemistry 

Organoids were washed thrice with 0.1 M phosphate buffer and fixed in freshly thawn 4% paraformaldehyde (Merck in phosphate buffer for 4 h at 4 °C. Fixed organoids were cryopreserved in 30% saccharose (VWR) in phosphate buffer at +4 °C overnight. After which, they were embedded in O.C.T. Compound (# 4583, Sakura, Tokyo, Japan) in 1 × 1 × 0.5 cm plastic moulds (, Andwin Scientific, Simi Valley, CA, USA) by incubating at RT for 20 min and freezing on a metal block at −70 °C. Organoids were then cryosectioned to 20 µm thickness on cryotome (CM1950, Leica, Wetzlar, Germany), sections were mounted on microscope glasses (Thermo Scientific) and stored in −70 °C. For heat-induced antigen retrieval, 10 mM sodium citrate (citrate (pH = 6.0, VWR) was heated to 92 °C, the slides were submerged and incubated for one hour, as the solution was allowed to cooled down in RT. Sections were blocked and permeabilized in 10% normal goat serum (Merck) and 0.05% Tween20 (Merck) in PBS for 1 h. Primary and secondary antibodies were diluted and incubated in the 5% normal goat serum in 0.05% Tween20 in PBS. Sections were incubated with primary antibodies (Table 4) at 4 °C overnight, washed 3× in 0.05% Tween20 in PBS and incubated with species-specific AlexaFluor secondary antibodies (1:500) for 1 h at RT. Slides were mounted in Vectashield mounting medium with or without DAPI (Vector Laboratories, Burlingame, CA, USA) for imaging. Staining controls that omitted the primary antibody were used to confirm staining specificity. 

### 2.7. Fluorescent Microscopy

All confocal images were obtained with a Zeiss Axio Observer inverted microscope (10×, 20×, 40× (oil) or 63× (oil) -objectives) equipped with LSM800 confocal module (Carl Zeiss Microimaging GmbH). DAPI and secondary antibodies were imaged with 405 nm (λex 353 nm/λem 465 nm), 488 nm (λex 495 nm/λem 519 nm) and 561 nm (λex 543 nm/λem 567 nm) lasers, respectively.

For imaging of representative figures, we used Leica Thunder 3D tissue imager, Figure 1b, (Leica microsystems CMS GmbH, Wetzlar, Germany), 10× objective using the Leica X application suite version 3.7.2.22383. The Zeiss confocal with 20× objective for Figure 1c–f, 40× for Figure 1g and 63× for Figures 1i, 2, Appendix A.

For quantification of morphology, we employed two imaging devices to acquire whole slice Z-stack tiles, capturing the entire Z-dimensional morphology of the microglia. We imaged the day 120 organoids at the Zeiss confocal, with seven focus planes (merged into maximal orthogonal projection), using the EC Plan-Neofluar 10×/0.30 M27 objective. Day 35 and day 66 organoids we imaged using 3D HISTECH Pannoramic 250 Flash III with 20× objective (3DHISTECH Ltd., Budapest, Hungary) and the extended view images exported to TIF for downstream analysis using the Caseviever version 2.4.

### 2.8. Quantification of Immunohistochemistry

The analysis on microglia morphology was done with the original CZI files (Zeiss LSM800 confocal), or the TIF files (3D HISTECH Pannoramic 250 Flash III). Skeletal analysis was performed using Fiji [73,74]. For the AI-based detection, the images were further compressed to a wavelet file format (Enhanced Compressed Wavelet, ECW, ER Mapper, Intergraph, Atlanta, GA, USA) with a target compression ratio of 1:5. The compressed virtual slides were uploaded to a whole-slide image management server (Aiforia Technologies Oy, Helsinki, Finland). A convolutional neural network-based algorithm was trained to detect ramified, intermediate, rod and spherical cells using cloud-based software (Aiforia version 4.8, Aiforia Create, Aiforia Technologies Oy, Helsinki, Finland). Final results were obtained via Origin 2019b (OriginLab, Northampton, MA, USA). 

For quantification of PSD95 puncta, we manually sampled four optical fields per organoid, four organoids per (+)/(−)ORG group, and captured a Z-stack of ten 109.38 µm × 109.38 µm (1580 px × 1580 px) focus planes with 0.28 µm interval, using the Plan-Apochromat 63×/1.40 oil DIC M27 objective. We performed counting of the PSD95 puncta with a set of 12 prominence thresholds, using the Find Maxima algorithm in Fiji [74]. We applied the same imaging parameters for assessing interactions between microglia and synapses, with maximum orthogonal projections acquired at ZEN 2.3 Blue edition or Zen 3.2 lite edition for the representative images in Appendix A.

### 2.9. Brain Slice Preparation for Electrophysiology

Brain organoids (107–213 days in culture) were embedded in 2% low melting point agarose and sliced (for whole-cell recordings, 350 μm; for multielectrode recordings, 500 μm) using a vibratome (model 7000 smz, Campden Instruments, Loughborough, UK) in chilled (4 °C), fully carbonated (95%/5%, O_2_/CO_2_) aCSF of the following composition (in mM). 92 N-methyl-D-Glucamine, 2.5 KCl, 20 HEPES, 25 NaHCO_3_, 1.25 NaH_2_PO_4_, 3 Na-Pyruvate, 2 Thiourea, 5 Na-Ascorbate, 7 MgCl_2_, 0.5 CaCl_2_, 25 Glucose (pH adjusted to 7.3 with HCl 10 M). After cutting, slices were placed in a custom-made chamber and allowed first to recover at 34 °C for 30 min in the recording solution (see below for composition, supplemented with 3 Na-Pyruvate, 2 Thiourea and 5 Na-Ascorbate) and then for 60 min in the same solution at room temperature (20–22 °C) before use for electrophysiology.

### 2.10. Whole-Cell Electrophysiology

A single slice (350 μm) was transferred and secured with a slice anchor in a large volume bath under an Olympus BX50WI microscope (Olympus corporation, Tokyo, Japan) equipped with differential interference contrast (DIC) optics, an ×40 water immersion objective and a charge-coupled device (CCD) camera (Retiga R1, Q-imaging, Teledyne Photometrics, Tucson, AZ, USA)). Slices were continuously perfused at a rate of 2.5–3 mL/min with a recording solution of the following composition (in mM): 120 NaCl, 2.5 KCl, 25 NaHCO_3_, 1.25 NaH_2_PO_4_, 2 CaCl_2_, 1 MgCl_2_, 25 Glucose. Organoids from both phenotypes were examined on a single experimental day with a blind protocol to the experimenter. Whole-cell current and voltage-clamp recordings were conducted at 32–33 °C with an Axopatch-200B amplifier (Molecular Devices, San Jose, CA, USA) using 5–8 MOhm glass electrodes filled with an internal solution containing (in mM) 135 Potassium-gluconate, 5 NaCl, 10 HEPES, 2 MgCl_2_, 1 EGTA, 2 Mg-ATP, 0.25 Na-GTP and 0.5% Biocytin (pH adjusted to 7.3 with osmolarity at 275–285 mOsm/L). Electrophysiological data were low pass filtered at 1 KHz (4-pole Bessel filter) then captured at 10 KHz via a Digidata 1440A A/D board to a personal computer, displayed in Clampex software (version 10.7, Molecular Devices) and stored to disk for further analysis. Single-cell electrophysiological current and voltage-clamp data were analysed in Clampfit (Molecular Devices). Synaptic data were detected and analysed with Mini Analysis software (Synaptosoft Inc., Decatur, GA, USA).

### 2.11. 60-3D-Multielectrode Electrophysiology

A single slice (500 μm) was transferred and secured with a slice anchor in the chamber of a 60-3D-multielectrode recording array (3D-MEA, Multichannel Systems-MCS, Reutlingen, Germany) with an 8 × 8 configuration (electrode impedance, 100–200 KOhm; spacing, 250 μm; height, 100 μm, conducting area only on top 20 μm) made of titanium nitride (TiN) and insulated with a thin layer of silicon nitride. Recordings were made with a MEA2100-Mini headstage (MCS) under a Leica S APO microscope equipped with a transillumination base and a digital camera (model MC170HD, Leica Microsystems). Slices were continuously perfused at a rate of 3–3.5 mL/min at 32–33 °C with the same recording solution as with whole-cell recordings and were allowed at least 20 min to settle in the MEA before any recordings or pharmacology was attempted. Organoids from both phenotypes were examined on a single experimental day with a blind protocol to the experimenter. Electrophysiological data were band passed between 300 and 3000 Hz (2nd order Butterworth) and were captured at 20 KHz via MCS-IFB 3.0 multiboot (MCS) to a personal computer, displayed in MCS Experimenter software (version 2.15) and stored to disk for further analysis. We used a firing algorithm based on standard deviation (5.5 times the baseline) of the unsmoothed basal firing rate histogram (in 0.1 s bins) and a minimum definition of 2 spikes per burst to detect bursts of spikes in spontaneously firing and NMDA responding channels. Local field potentials (LFPs) from NMDA responding channels of interest (COIs), were extracted from the raw signal using a 200 Hz low pass filter (2nd order Butterworth), were stored in a separate file and were subjected to fast Fourier transformation (FFT) and spectrogram visualisation (Welch windowing, 50% overlapping window) and power spectral density (PSD, normalized as log PSD, in dB) analysis for different frequency bands (delta [δ] 0.5–3 Hz, theta [θ] 4–10 Hz and gamma [γ] 30–100 Hz) for the duration of the recording (in 10 s bins). PSD was also computed in 2-min epochs for the frequency range between 1 to 100 Hz (frequency bins of 0.61 Hz) at baseline, during NMDA application and during wash-out and the mean relative power induced by drug application was calculated by subtraction from the baseline epoch for each individual electrode. A three-bin wide (each 10 s) Boxcar and Gaussian smoothing routine was used for firing/bursting rate and spectral power determinations, respectively. Firing, bursting and LFP PSD analysis of data was conducted in Neuroexplorer software (Plexon Inc, Dallas, TX, USA) while spike sorting for single unit analysis (through principal component analysis) was conducted in spike2 software (Cambridge Electronic Design, Cambridge, UK).

### 2.12. Statistical Analysis

All data represent mean  ±  s.e.m. The morphologies of microglia were analysed on Origin (OriginLab Corporation, Northampton, UK) with significance tested using the Tukey test. All other statistical tests were performed on Prism (GraphPad Software, La Jolla, CA, USA), with two-way-ANOVA for time plot comparisons of electrophysiological data, Figure 5j,k, and non-parametric Mann–Whitney unpaired tests for PSD95 puncta and electrophysiology results (except for paired *t*-test between (+)ORG and (−)ORG + NMDA in Figure 4h). The particulars of appropriate statistical tests used are detailed in text.

## 3. Results

### 3.1. Erythromyeloid Progenitors Migrate into Brain Organoids and Mature into IBA1+ Cells

Given that microglia migrate into brain early during embryogenesis, we hypothesized that erythromyeloid progenitors (EMPs) would migrate spontaneously to brain organoids at an early state, as had earlier been shown for more mature induced pluripotent stem cell (iPSC)-derived microglia [54,69]. To test this idea, we differentiated iPSC-derived erythromyeloid progenitors (EMPs) according to previously published microglial differentiation protocol [69]. We incorporated them in day 30 cerebral brain organoids generated using an undirected protocol similar to previously published work [20]. The schematic illustration in Figure 1a outlines the merging of the two protocols. The incorporated EMPs establish an IBA1+ population and colonise the organoid, as presented in Figure 1b. The robustness of differentiation was confirmed by performing differentiation using four independent and healthy iPSC lines (Table 1). Immunohistochemical stainings against the EMP marker CD41 revealed the presence of the EMP population suspended in Matrigel adjacent to the organoid tissue still five days after the incorporation, at 35 days in vitro (DIV) (Figure 1c). Cells positive for the microglia marker IBA1 and negative for CD41 had begun to infiltrate the nascent cortical loops, as identified by their tissue architecture shown in immunohistochemistry (IHC) staining against TBR2+ intermediate progenitors and DCX+ young neurons (Figure 1d). Double-staining against IBA1 and the microglia/macrophage specific transcription factor PU.1 revealed the microglia-like cells to be positive for both markers (Appendix A) [75]. Moreover, IBA1+ cells attained less spherical morphology upon arriving at the organoid tissue, and as the organoids matured, the IBA1+ cells portrayed a spectrum of morphologies (Figure 1e,i, 120 DIV). The organoids showed radial maturation of the cortical plate (Figure 1f, 120 DIV), as shown previously [20,39] and the IBA1+ cells exhibited tangling with neuronal processes (Figure 1f, d66 DIV). Furthermore, in accordance with earlier work, we observed minor native populations of microglia in organoids without incorporated microglia progenitors [56]. However, counting the IBA1+ cells in day 120 organoids with and without incorporated microglia, abbreviated here (+)ORG and (−)ORG, respectively, showed significantly more microglia in the (+)ORG group (Appendix A).

As the microglia take increasingly complex morphologies during brain development in vivo [76], and given the breadth of variation in the morphologies of IBA1+ cells even within a single organoid, we sought to answer if the relative portions of particular morphologies within the IBA1+ cell population changes as the organoid matures. For this, we employed an artificial intelligence (AI) platform offered by Aiforia and trained the convolutional neural network-based algorithm to annotate four different morphology types (Figure 1i) from images of immunohistological stainings against IBA1 in organoid cryosections. We quantified the numbers of ramified, intermediate, rod-shaped and spheric cells in organoid sections at 35, 66 and 120 DIV (Figure 1h). At 35 DIV, the IBA1+ populations consisted mainly of cells detected as spheric and intermediate. At 66 DIV the portions of ramified and intermediate cells increased and spheric cell decreased. On day 120, the ramified cells were further enriched within the IBA1+ population, while the portion of other types did not significantly change from day 66. In parallel to the AI-based quantification, we performed skeletal analysis using ImageJ [73,74], entailing manual annotation of the cells and yielding per-organoid average counts of branches, junctions, endpoints and triple points (Appendix A). In agreement with the AI-based detection results, the average complexity of cell morphology increased from day 35 to 66. However, the complexity decreased from day 66 to day 120. Combining the results from the AI-based classification and skeletal analysis suggests that indeed, the IBA1+ cells attain more complex morphologies as they mature within the organoid environment. However, while the average morphological complexity decreased from day 66 to day 120, as shown by skeletal analysis, the portion of the ramified subset increases, as demonstrated by the AI-based detection.

**Figure 1 cells-11-00124-f001:**
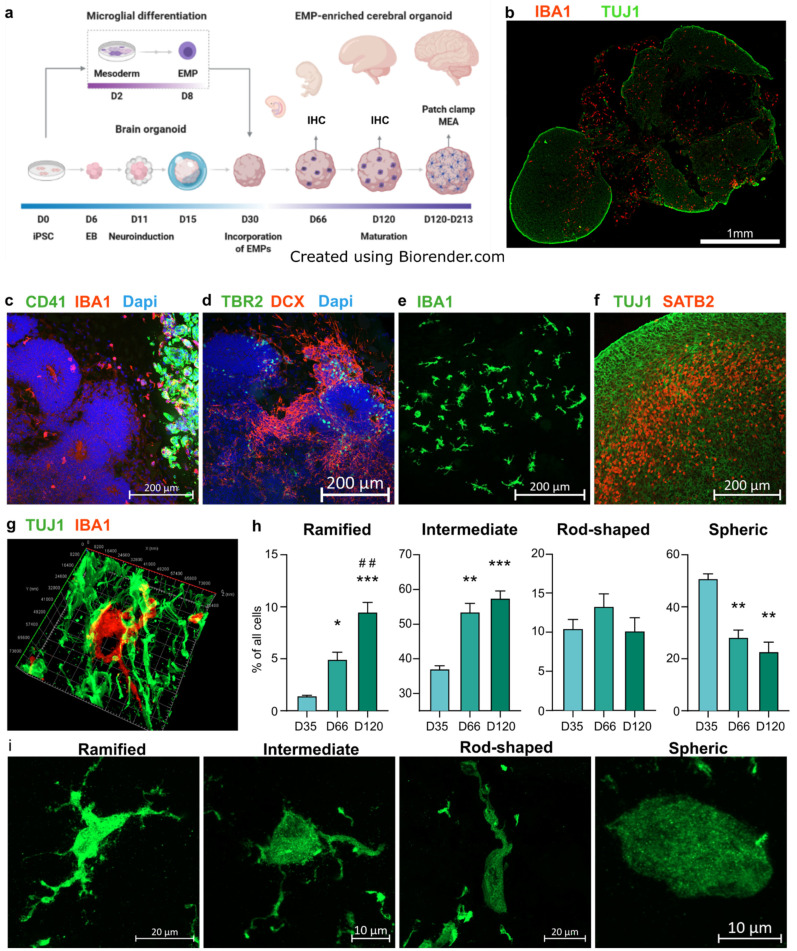
Erythromyeloid progenitors (EMPs) migrate to establish an IBA1+ population in cerebral organoids. (**a**) Schematic drawing depicting the introduction of EMPs to ORGs and subsequent experiments. (**b**) Immunohistological staining of IBA1+ cells colonizing a TUJ1 stained cORG at 120 days in vitro (DIV). (**c**,**d**) Immunohistological stainings of adjacent sections of an ORG 35 DIV, five days after incorporation of EMPs. TBR2+ and DCX+ cells show the formation of cortical loops, while CD41 shows the locus of incorporation and IBA1, the cells migrating from the locus to the vicinity of the nascent cortical loops. (**e**) A population of IBA1+ cells in taking different morphologies at 120 DIV. (**f**) Maturation of a cortical loop, as shown by immunostaining of SATB2 at 120 DIV. (**g**) A 3D reconstruction of an IBA1+ cell and its processes closely intertwined with TUJ1+ neurons at 66 DIV. (**h**) Detection and quantification of different morphology types of IBA1+ cells in organoids 35, 66 and 120 DIV, using the artificial intelligence platform by Aiforia. The Y-axis presents the portion of the respective morphology type in all the detected IBA1+ cells. The bars represent the mean (+/− SEM) percentage of three organoids for the 35 DIV group (one batch), six for 66 (DIV (one batch) and nine for 120 DIV (two batches), Tukey test, significance *** *p* < 0.001, ** *p* < 0.01, * *p* < 0.05 compared to 35 DIV, ## indicate similarly significance compared to 66 DIV; (**i**) Representative orthogonal projection images of all the quantified morphology types, imaged from a single section of an organoid at 120 DIV. All results presented here come from organoids of the iPCS line MBE2968, while the qualitative results in panels (**b**,**e**–**g**,**i**) was replicated using the other three lines, MAD6, BIONi010-C-2 and Ctrl8 c2.

### 3.2. IBA1+ Cells Interact with Pre- and Post-Synaptic Elements

Immunohistochemical staining against postsynaptic density protein 95 (PSD95) (Appendix A) showed an assortment of PSD95+ puncta of different sizes, in line with its physiological role as a dynamically regulated scaffolding protein in the dendritic spine [77,78,79,80], and the sizes of representative puncta corresponded to the sizes of PSD95 nanocluster detected in the human brain [81]. Moreover, while most PSD95+ puncta were scattered (Appendix A), representing the typical spatial arrangement shown earlier [81], distinct areas in the sections portrayed PSD95+ puncta arranged around nuclei (Appendix A), also shown in earlier reports [82,83], and possibly representing perisomatic synapses [84,85], with no apparent differences in PSD95 localisation between (+)ORG and (−)ORG groups. IHC against synaptophysin (SYP), a protein of synaptic vesicles and denotating presynaptic compartments [86], showed likewise intense punctate staining, though bigger puncta as in PSD95 IHC, and less intense staining localised to stripes resembling neuronal processes (Appendix A).

Inspired by the IHC staining against IBA1 and B-tubulin showing the IBA1+ cells to be intensively intertwined within neurons (Figure 1g), we further investigated physical proximity between the IBA1+ cells and synaptic material. Indeed, double staining for IBA1 and post-synaptic density protein 95 (PSD95), and IBA1 and the presynaptic synaptophysin (SYP) at 120 DIV revealed instances of synaptic material being inserted in an open pocket on the surface of the IBA1+ cells (Figure 2a,b). Orthogonal dissection images revealed SYP-stained material to fill the entire pocket, while PSD95 filled only part of the pocket. Additionally, for both SYP1 and PSD95, we observed partial co-localisation with IBA1 within the pocket. For SYP1, we also observed instances of non-pocketed co-localisation with a process of an IBA1+ cell. Furthermore, at 150 DIV, we observed PSD95+ material within an IBA1+ cell, wherein PSD95 existed as puncta and as diffuse spheres (Figure 2c). Taken together, the images compose immunohistologal evidence of the IBA1+ cells physically interacting with both pre- and post-synaptic particles and internalising post-synaptic material. 

Next, to analyse if the interaction between microglia and synapses was reflected in the numbers of PSD95 puncta, we quantified the numbers of PSD95 puncta in the (+)/(−) ORGs at 120 DIV using the FindMaxima algorithm in Fiji [74]. Given the impressive variance in puncta sizes, we applied a series of incrementally increasing prominence thresholds to interrogate the prevalence of different puncta sizes, but saw no significant difference in the PSD95 puncta counts between the (+)/(−) groups at any prominence threshold (Appendix A). 

**Figure 2 cells-11-00124-f002:**
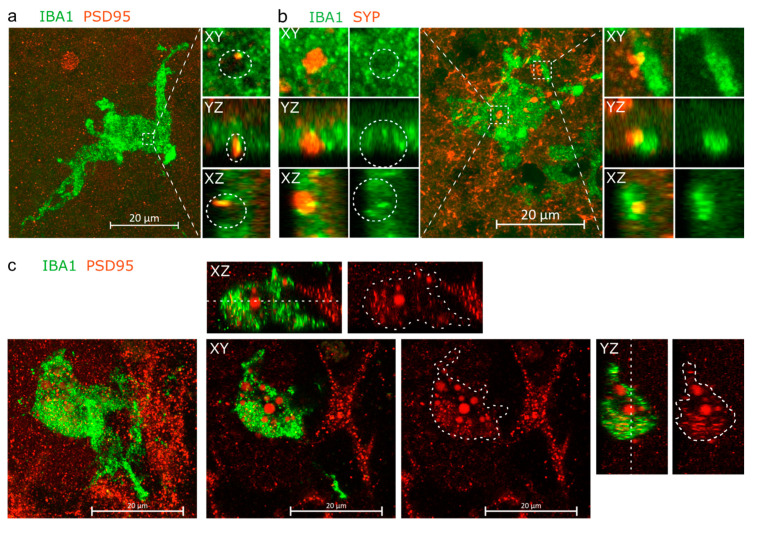
IBA1+ cells are in intimate proximity with neurons. (**a**) Orthogonal projection and orthogonal dissections of PSD95+ post-synaptic material embedded within a pocket on the surface of an IBA1+ cell. (**b**) Instances of SYP+ presynaptic material embedded within a pocket on the soma of IBA1+ cell and partially internalised by process of the cell. (**c**) Orthogonal projection, with the XY, XZ and YZ planes showing PSD95+ material located within the IBA1+ cell. The white cut line crossing the IBA1+ cell on the XZ and XZ planes marks the Z-coordinate of the XY plane, while cut line areas drawn on the red-channel images overlay perimeter of the IBA1+ cell. Panels a and b come from organoids of the iPCS line MAD6, while panel c comes from MBE2968.

### 3.3. IBA1+ Cells Expedite Neuronal Maturation in Cerebral Organoids 

To assess whether enrichment of organoids with IBA1+ cells has the capacity to alter single neuron properties in the developing organoids, we performed whole-cell recordings in acute organoid slices at days 107–120 and days 150–165. The recordings were carried out under voltage-clamp conditions to measure the maximal sodium and potassium (holding voltage—70 mV, test voltage −20 mV and +10 mV, respectively) and passive leak current density (holding voltage—50 mV, test voltage −120 mV) as markers of neuronal maturation and stage of development. Currents recorded were normalised to cell size and expressed as pA current per pF of cell capacitance. We detected a statistically larger potassium current density in (+)ORG at the early time point and a significantly larger mean sodium current density in neurons in (+)ORG at both recorded time points (Figure 3a,b). No changes were seen in the background mean maximal leak current density between the two groups (Figure 3a,b) or basal electrophysiological parameters of the neurons measured under current-clamp conditions (input resistance, resting membrane potential or membrane time constant, Figure 3c). We found that the increased sodium current density translated functionally to a significantly larger prevalence of repetitive action potential (AP) firing neurons in (+)ORG (Figure 3d–f) and a larger, albeit marginally insignificant, prevalence of neurons firing non-adaptively during the current steps (Figure 3g). Moreover, neurons from (+)ORG were more likely to express active ionic currents and, in particular, a presumed low-threshold calcium current often leading to a low threshold potential-spike (LTS), following release from a prolonged hyperpolarisation (Figure 3h,i). Furthermore, we only detected spontaneous excitatory post-synaptic currents (sEPSCs) under voltage-clamp in neurons in (+)ORG but not in (−)ORG (Figure 3j–l).

### 3.4. IBA1+ Cells Mediate Emergence of Spontaneous Bursting Activity in Cerebral Organoids

Next, we looked if microglia would boost the emergence of network activity, previously reported in organoids [34]. 3D-MEA electrophysiological recordings were undertaken from acute slices of day 150–165 organoids to examine the developing characteristics of the neuronal network properties. We detected only a single active electrode track with spontaneously occurring multiunit activity from (−)ORG slices, which was in striking contrast to the (+)ORG slices where we detected 24 spontaneously active electrode tracks under basal conditions (Figure 4a,b). (+)ORG slices exhibited activity patterns in active electrode tracks ranging from irregular firing to regular bursting (Figure 4b–d). Such bursting activity could be often discriminated successfully, through principal component analysis, into single unit activity (SUA; Figure 4c) from presumed single neuron firing, but each unit appeared relatively asynchronous in bursting phase to others and thus mean population spiking plots remained flat (Figure 4e) and did not follow any regular oscillatory activity with typical high amplitude peak spiking-low amplitude silent pause features expected from synchronously bursting networks as described before by others [34]. The differences in the prevalence of activity and functional characteristics were not secondary to slice-MEA placement or viability of the neurons as we could reverse the silent electrode tracks of the (−)ORG slices to active by exciting the neurons with the application of NMDA, resulting in the emergence of brief bursting activity (Figure 4a,h). Whole-cell recordings from organoids that have undergone MEA recordings confirmed the presence of bursting neurons from resting (0 pA) or near resting membrane potential (from +2 to +10 pA current injection, Figure 4g) in slices from (+)ORG but rarely from slices of (−)ORG (Figure 4f). 

To further elaborate on our functional findings and to ascertain whether the striking differences seen would persist in time, we examined four organoid-slices per group (taken from two different batches) at DIV 200–213 using 3D-MEA techniques coupled to NMDA pharmacology to induce brief periods of persistent neuronal bursting and local field potential (LFPs) oscillations. Unlike our previous results at DIV 150–165, we found a similar number of spontaneously active channels per organoid-slice under both baseline and under induced conditions via NMDA receptor activation (baseline active channels, Figure 5a). We also found only a small minority (circa 5–10%) of spontaneous active channels that exhibited signs of bursting in baseline condition in both phenotypes (Figure 5b) unlike the high burst prevalence in (+)ORG only at DIV 150–165 (Figure 4b) suggesting that time spent in culture reduced the incidence of finding spontaneous bursting channels in our organoids. To understand the developing dynamics of the network, we focused our analysis on the impact of brief NMDA receptor activation (2 min) on spike firing and spike bursting characteristics during our pharmacological experiments. The mean firing rate during NMDA perfusion (2-min average) was significantly higher in (+)ORG phenotype (Figure 5c,d). Burst firing induced by NMDA was significantly higher in (+)ORG phenotype (Figure 5e,f), while we found significantly more bursting channels induced by NMDA superfusion in (+)ORG than in (−)ORG phenotypes (Figure 5g). Further analysis of the NMDA-induced bursting characteristics revealed that (+)ORG exhibited significantly higher, mean and mean peak intra-burst firing frequency (Figure 5h). To probe further the characteristics of the developing network’s behaviour in organoids, we examined NMDA-induced oscillatory activity in different frequency bands by analysing LFPs from NMDA responding channels (*n* = 44 vs. 66 responding channels for (+)ORG and (−)ORG phenotypes, Figure 5i–k). Power spectral density (PSD) analysis revealed that NMDA induced a strong, but differential enhancement of specific frequencies, especially in the high frequency γ-band (30–100 Hz) range (Figure 5i). We found that PSD induced by NMDA was significantly different between (+)ORG and (−)ORG phenotypes (Figure 5j). In particular, the power of the induced γ-band component of the LFP was highly significant and larger in (+)ORG than (−)ORG slices (Figure 5k). 

## 4. Discussion

Earlier work has characterised the development of neuronal functions in cerebral organoid in single-cell sequencing and functional recordings, and thus consolidated cerebral organoids as a platform that not only recapitulates the tissue architecture of the developing cortex, but also developmental trajectories of maturing neurons and their electrophysiological properties [19,34,35,37]. In parallel, recent publications report several strategies of including microglia in the organoids, making it a radically more complete platform to model CNS in vitro [54,55,56,58,65]. These two lines of exploration convened in a very recent report that benchmarked the impact of incorporated primary microglia on the transcriptome and electrophysiological properties of cerebral organoids [65]. 

Here, we push the exploration further, and evaluate at multiple developmental time points the functional characteristics of single neurons and multichannel network properties of cerebral organoids that have co-developed with microglia-like cells.

Our approach of incorporating erythromyeloid progenitors that mature into microglia-like cells within the organoid environment, similar to previous reports [57,58], strike a compromise between the methods reported so far; that of incorporating mature iPCS-derived microglia [54,55,69], adjusting the organoid protocol to give rise native microglia [56], and incorporating primary microglia received from mid-gestation human brain [65]. Each approach enjoys their benefits. Inducing the emergence of native microglia is more likely to align the co-development of microglia and neurons closer to the synchrony observed in vivo development [87], while exogenous incorporation efforts need to determine the right development stages for incorporation to achieve the same. However, exogenous microglia allow for convenient employment of microglia specific reporter genes or mutations, and assessment of the impact of microglia to the organoid functions. Incorporating primary microglia delivers a benchmark for transcriptome signature and functional analysis with high in vivo relevance, while being methodologically far less accessible than iPCS-derived cells. 

Our findings of physical contact between microglia-like cells and synaptic components are in line with previous work [55,56,58,65], and encourages future efforts in live-imaging of microglia-neuron contacts to determine if the IBA1+PSD95 contact is transient or if it results in internalisation of synaptic components [88]. Such experiments would effectively dissect the question whether a particular contact is supporting or degrading the synapse [44,51,88,89]. 

Beyond the close interaction of microglia and synapses, the functional synaptic/network differences seen here between different type of organoids demonstrate a generalized more mature and diverse neuronal phenotype to emerge in organoids with microglia. We recorded both single neurons using whole-cell recordings and spontaneous single and multiunit neural activity using extracellular electrophysiology with multi-electrode arrays, and found an increased prevalence of single neurons with bursting behaviour in (+)ORGs compared to (−)ORGs consistent with the presumed role of microglia in the development of neuronal properties and network formation [50,90]. Our data suggest that neurons from (+)ORG exhibit clear signs of strong and regular, spontaneous and NMDA-induced single-cell bursting characteristics without, however, having developed yet concrete network features such as synchronised population activity [37]. Highly synchronous NMDA-induced bursting and high power γ-band oscillations in LFPs represent further evidence that organoids with microglia can attain more robust properties of physiologically relevant cortical behaviour that requires an intact wired network of neurons, amongst them often interneurons. Although such properties are not completely absent in (−)ORGs, suggesting that (−)ORGs used herein have the tendency to structurally wire into functional networks, microglia-like cells affect the emergence and development of spontaneous and induced network properties by bringing and maintaining them in more dynamic maturational state at an earlier time point. The larger sodium-potassium current density, the more consistent spontaneous and induced firing and bursting characteristics, the presence of greater ion channel diversity, the emergence of synaptic activity as well as the greater power of oscillatory phenomena in slices of (+)ORGs suggest that microglial enrichment has affected the development and the fundamental aspects of maturation of single-cell and network properties of neurons. Admittedly, whole-cell recordings undertaken herein are laborious and have an inherent lower yield than MEA recordings, but they complement the MEA data and shed light on the intricate development of intracellular properties reported here. These data combined show that single-cell sodium and potassium current properties, that are taken as a sign of neuronal maturation [91] are more developed in the neurons in the (+)ORGs which lends a mechanistic insight into the observed differential behaviour (e.g., firing and bursting propensity) of. the two phenotypes of organoids screened in our study. The results obtained herein in slices are in general good agreement with the recent study from Popova and colleagues [65] where late and short-lived incorporation of microglia in organoids resulted in enhanced network activity judged by functional MEA recordings. Our results extend this view and support the notion that the regulation of sodium and potassium channel development by microglial-like cells give rise to more mature neuronal functions that ultimately manifest as increased neuronal network function.

The striking differences in single-cell electrophysiological properties between (+) and (−)ORGs prompt for dissecting the temporal dynamics of the co-development by depletion of microglia. This might help in directing further microglia-neuron interaction studies to the most relevant stages of organoid development. In addition, single-cell sequencing studies will reveal the neuronal populations that are emerging in response to incorporation of microglia-like cells, similarly as previous sequencing work has revealed enrichment of GABAergic interneurons as the organoids mature [37].

Detailed interrogation of the microglia-neuron interaction is mandatory for understanding the aspects of neuronal development, and more importantly, the aspects of neurodevelopmental disorders most amenable to modelling by using the current organoid approaches [92]. However, a key consideration for organoid-based models is to minimise the emergence of the hypoxia-derived necrotic core as the organoid grows in size [59]. This is an important consideration in cerebral organoids, as the introduction of vasculature was shown to boost neuronal development [59,60,62], and especially important for studying microglia-like cells, as in vivo microglia react to tissue damage [93]. Future work will need to employ organoids that have minimal or no necrosis, by committing to an air-liquid-interphase approach, organoids with vasculature, or to a microfluidic system [55,59,60,62,94].

While our work here focuses solely on analysing neuronal functions arising due to presence of microglia-like cells, future key aspects in characterising microglia-containing organoids would be to interrogate the identity and behaviour of the microglia-like cells within. CD11b sorting and single-cell sequencing allow for determining how closely the iPSC-EMP-derived microglia-like cells correspond to the in vivo microglia or to the primary microglia maturing in organoids [65]. While previous work with microglia-containing organoids has shown the microglia to respond to inflammatory insults [55,56,58,65], an intriguing next step is to analyse if the response results in divergent “non-homeostatic” or “disease associated” subpopulations of microglia, as has been shown for example for Alzheimer’s disease or multiple sclerosis [95,96,97]. Microglia-containing organoids serve as a relevant platform to model microglia responses in different brain diseases.

## Figures and Tables

**Figure 3 cells-11-00124-f003:**
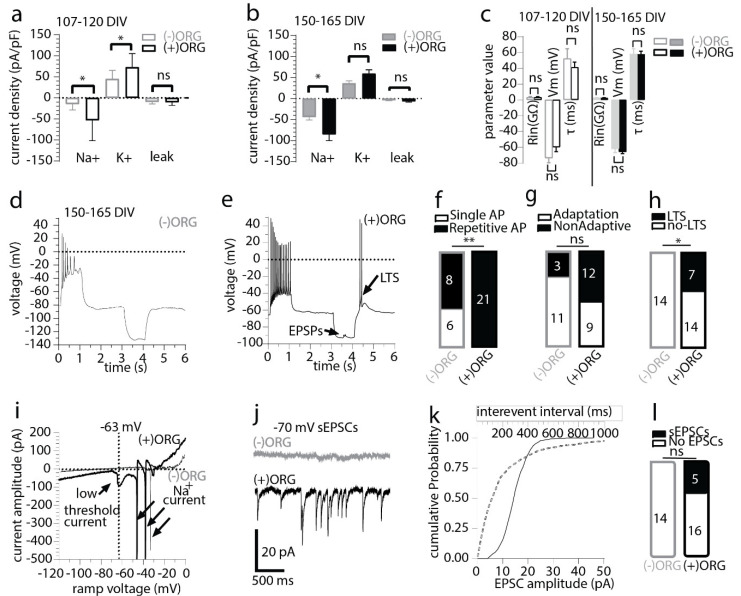
Incorporated IBA1+ population supports neuronal maturation and synaptic signalling in cerebral organoids. (**a**) Maximal sodium, potassium, and leak, current density in 107–120 day old organoids. *n* = 8 vs. 9 neurons (mean ± SEM; Mann–Whitney unpaired test, * *p* < 0.05, ns-non-significant *p* > 0.05, 4 to 5 organoids per group from two independent batches). (**b**) Maximal sodium, potassium and leak current density in 150–165 day old organoid. *n* = 14 vs. 21 neurons (mean ± SEM; Mann–Whitney unpaired test, * *p* < 0.05, ns *p* > 0.05, 6 to 9 organoids per group, from three independent batches). (**c**) The basal (resting membrane potential-Vm) and evoked electrophysiological properties (during a −10 to −20pA 1 s hyperpolarising pulse under current-clamp mode, input resistance-Rin, membrane time constant- τ) did not differ at either developmental time point between the two organoid groups. *n* = 8 vs. 9 neurons and *n* = 14 vs. 21 neurons as detailed in a and b (mean ± SEM; for all comparisons, Mann–Whitney unpaired test, ns *p* > 0.05). (**d**) Electrophysiological traces recorded in current-clamp mode showing the responses of an (−)ORG neuron to positive and negative current injections (±12.5 pA, 1 s). Note the low AP amplitude, complete cessation of firing at the end of the depolarising step and the pure passive responses during and after the hyperpolarising step. (**e**) Electrophysiological traces recorded in current-clamp mode showing the responses of an (+)ORG neuron to positive and negative current injections (±20 pA, 1 s). Note the large AP amplitude, the sustained firing at the end of the depolarising step and the low threshold spike (LTS) after the end of the hyperpolarising step. (**f**) Summary results for single AP and repetitive AP firing neurons recorded in (−)ORG and (+)ORG slices (Chi-square with Yate’s correction test, ** *p* < 0.01). (**g**) Summary results for spike frequency adaptation during the 1 s depolarizing step recorded in (−)ORG and (+)ORG neurons (Chi-square with Yate’s correction test, *p* > 0.05). (**h**) Summary results for the presence of post-hyperpolarisation low threshold spike-potential (LTS/LTP) (Chi-square with Yate’s correction test, *p* < 0.05). (**i**) Electrophysiological traces recorded with a voltage ramp (speed 0.14 V/s) taken from neurons shown in d and e, showing an inward current developing in the subthreshold range, which is responsible for the LTS responses recorded in the (+)ORG neuron. Note the complete absence of current nonlinearities in the (−)ORG neuron in subthreshold voltage potentials. (**j**) Electrophysiological traces of spontaneous excitatory postsynaptic currents (sEPSCs) recorded under voltage clamp (holding potential of −70 mV) from an (−)ORG and (+)ORG neuron. Note the complete absence of sEPSCs in the (−)ORG neuron. (**k**) Cumulative distribution curves for inter-event interval (dotted gray line) and amplitude (solid black line) of sEPSCs detected on the (+)ORG neuron presented in j. (**l**) Summary results for the presence of sEPSCs in (−)ORG and (+)ORG neurons (Chi-square with Yate’s correction test, *p* > 0.05). All recording made from organoids of the iPCS line MBE2968.

**Figure 4 cells-11-00124-f004:**
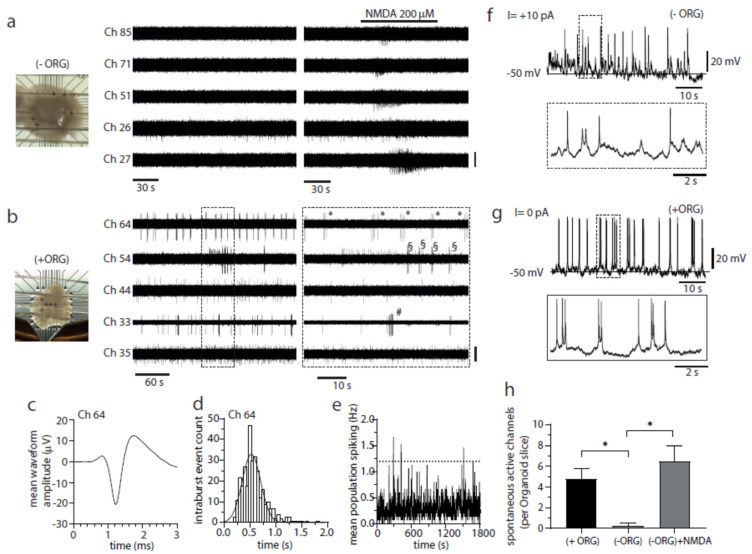
Incorporated IBA1+ population promotes neuronal bursting and network activity in cerebral organoids. (**a**) Image of an (−)ORG slice recorded with a 3D 8 × 8 MEA (left) and raw voltage recordings of spiking activity from five representative electrodes (depicted as + symbols on image) before and after NMDA perfusion (middle and right panels). Only channel 26 exhibited activity in baseline conditions (one active electrode detected from 4 organoid slices), while NMDA perfusion effectively and quickly elicited a brief burst of activity on all five channels (bar, 20 µV). (**b**) As described above in a but for a (+)ORG slice. The regular bursting activity of 3–7 spikes was detected in three electrodes (33, 54, 64) (twenty-four active electrodes detected from 5 -organoid slices). Symbols (*, # and §) mark individual bursts in each different channel (bar, 20 µV, except for Ch 33, 40 µV). (**c**) Mean spike waveform after spike sorting (255 spikes) using principal component analysis from raw signal depicted in b from channel 64. A single tight cluster with a well-defined AP waveform was detected consistent with the recording being from a single neuron (single unit activity, SUA). (**d**) Intra-burst spike interval histogram (50 ms bins) for SUA from channel 64 presented in (b,c). Ten of total 24 active electrode tracks detected from 5 (+)ORG slice-organoids exhibited signs of bursting. The single Gaussian function fitted to the data is centred at 524 ± 178 ms (mean ± SD) while the mean inter-burst interval for this neuron was 10.7 ± 4.2 s (mean ± SD). (**e**) Mean population spike activity from all the active electrodes the (+)ORG slice presented in b (in 1 s bins). Although individual electrodes exhibited bursting activity, this activity was largely asynchronous between electrodes, as seen by the typical flattened population spiking response over the time course of 10 min (dotted line represents network burst threshold set at 6 times the SD of the baseline). (**f**) Electrophysiological traces from whole-cell recording from a (−)ORG neuron exhibiting a current-induced (+10 pA) low amplitude, short term bursting behaviour. (**g**) Electrophysiological traces from whole-cell recording from a (+)ORG neuron exhibiting high amplitude, robust, regular bursting behaviour from its resting membrane potential. (**h**) Bar chart comparison of active electrode tracks recorded per organoid slice. (+)ORG slices exhibited a statistically significant larger number of detected active electrode tracks compared to (−)ORG slices. This lack of spontaneous activity could be reversed by perfusion of NMDA that caused the emergence of transient activity on previously silent electrodes recorded from (−)ORG slices (1 vs. 24 active electrodes, *n* = 4 (−)ORGs and *n* = 5 (+)ORGs, respectively, from two independent batches). All recording were made from organoids of the iPCS line MBE2968.

**Figure 5 cells-11-00124-f005:**
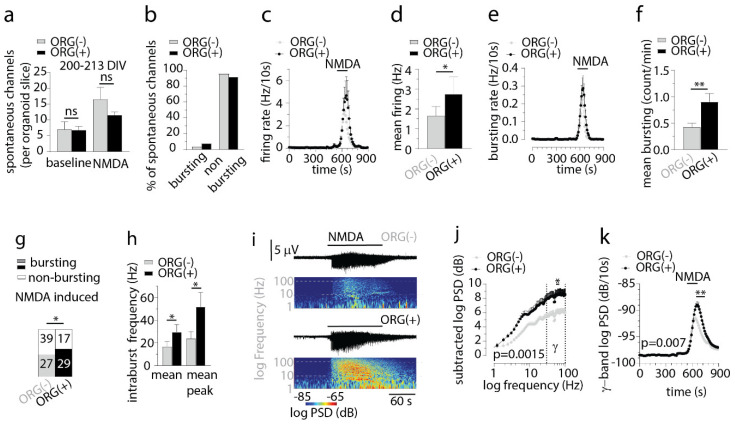
Incorporated IBA1+ population affects NMDA-induced firing and bursting characteristics and the development of high frequency oscillations. (**a**) Bar chart comparison of spontaneous active channels per organoid-slice in (+)ORG and (−)ORG slices at 200–213 DIV. No differences were seen between the two phenotypes in the number of either spontaneous or NMDA-induced active channels (Mann–Whitney unpaired test, ns *p* > 0.05, from 4 vs. 4 organoid-slices from (+)ORG and (−)ORG phenotype). (**b**) Bar chart comparison of bursting prevalence of spontaneously active channels in baseline. About 5–10 % of spontaneously firing channels exhibited bursting behaviour in baseline in both phenotypes. (**c**) Plot of spike firing rate (in 10 s bins) for the time course of a pharmacological experiment with NMDA in (+)ORG and (−)ORG slices (*n* = 46 vs. 66 NMDA responding electrodes, respectively, unsorted multiunit activity). (**d**) Bar chart comparing average two-minute firing rate during NMDA superfusion. (+)ORG slices exhibited a statistically significant higher firing frequency during NMDA superfusion (Mann–Whitney unpaired test, * *p* < 0.05). (**e**) Plot of spike bursting rate (in 10 s bins) for the time course of a pharmacological experiment with NMDA in (+)ORG and (−)ORG slices (*n* = 46 vs. 66 NMDA responding electrodes, respectively, unsorted multiunit activity). (**f**) Bar chart comparing average two-minute bursting rate during NMDA superfusion. (+)ORG slices exhibited a statistically significant higher bursting frequency during NMDA superfusion (Mann–Whitney unpaired test, ** *p* < 0.01). (**g**) Bar chart comparison of numbers of channels responding to NMDA with burst firing. (+)ORG slices exhibited a statistically significant higher prevalence of exhibiting bursting sequences during NMDA superfusion (Chi-square with Yate’s correction exact test, * *p* < 0.05). (**h**) Bar chart comparison of mean intra-burst and mean peak intraburst frequency during NMDA superfusion. (+)ORG slices exhibited statistically significant higher mean and mean peak intra-burst frequency (both Mann–Whitney unpaired test, * *p* < 0.05). (**i**) Spectrograms from local field potentials (LFPs) showing power spectral density (PSD) at different frequencies against time for a pharmacological experiment with NMDA in (+)ORG and (−)ORGslices. Note the persistent high power of γ-frequency (30–100 Hz) LFP activity emerging upon NMDA superfusion in (+)ORG slices in comparison to (−)ORG slices. (**j**) Subtracted log PSD of LFPs against frequency (in 0.61 Hz bins) for (+)ORG and (−)ORG slices before and during the two-minute pharmacological exposure to NMDA. The effects of NMDA on PSD at different frequencies were highly significant (two-way ANOVA, *p* < 0.01, with Bonferroni multiple post-tests against frequency at indicated points in γ-frequency range, * *p* < 0.05). (**k**) PSD of LFPs for γ-band frequency (in 10 s bins) for (+)ORG and (−)ORG slices during the time course of a pharmacological experiment with NMDA. NMDA caused a highly significant differential power induction in the γ-frequency in (+)ORG slices compared to (−)ORG slices (two-way ANOVA, *p* < 0.01, with Bonferroni multiple post-tests against time at indicated points in, ** *p* < 0.01). Recording were made from organoids of the iPCS lines BIONi010-C-2., and Ctrl 8 C2., both lines having one independent batch.

**Table 1 cells-11-00124-t001:** Lines of induced pluripotent cell used in the presented work.

Cell Line	MBE2968 c1	Mad6	BIONi010-C-2	Ctrl 8 c2
**Gender**	F	M	M	F
**Health status**	Healthy	Healthy	Healthy	Healthy
**Age, years**	65	63	15–19	Adult
**APOE type**	ε3/ε3	ε3/ε3	ε3/ε3	ε3/ε3
**Sample origin**	Skin biopsy	Skin biopsy	Skin biopsy	Skin biopsy
**Reprogramming method**	Episomal nucleofaction	Sendai virus	Non-integrating episomal	Sendai virus
**Karyotype**	46XX	46XY	46XY	46XX
**Reference**	[69]	-	[70,71]	[72]
**Used in experiments**	Figure 1 Figure 2c Figure 3 Figure 4, Appendix A	Figure 1b,e–g,i Figure 2a,b Appendix A Appendix A Appendix A	Figure 1b,e–g,i Figure 5	Figure 1b,e–g,i Figure 5

**Table 2 cells-11-00124-t002:** qPCR probes used to characterize the induced pluripotent cell line Mad6.

Purpose	Target	Company and Cat#
Pluripotency markers (qPCR)	*Nanog*	Thermo Fisher Scientific, Hs02387400 g1
	*Lin28*	Thermo Fisher Scientific, Hs00702808 s1
	*Sox2*	Thermo Fisher Scientific, Hs01053049 s1
House-keeping genes (qPCR)	*ACTB*	Thermo Fisher Scientific, 4326315E
Sendai virus	SeV	Thermo Fisher Scientific, Mr04269880_mr

**Table 3 cells-11-00124-t003:** Antibodies used to characterize the induced pluripotent cell line Mad6. Antibodies produced by R&D Systems, Minneapolis, MN, USA, Merck KGaA, Darmstadt, Germany and Thermo Fisher Scientific.

Purpose	Antibody	Dilution	Company and Cat#
Differentiation markers	SOX17 PE (Endoderm)	As per datasheet	R&D ICI9241P
	OTX2 AF488 (Ectoderm)	As per datasheet	R&D ICI979G
	BRACHYURY APC (Mesoderm)	As per datasheet	R&D IC2085A
Pluripotency markers	Mouse anti-OCT4	1:400	Merck, MAB4401
	Goat anti-NANOG	1:100	R&D Systems, AF1997
	Mouse anti-SSEA4	1:400	Merck, MAB4304
	Mouse anti-TRA-1-81	1:200	Merck, 4381
Secondary antibodies	Goat anti-mouse Alexa Fluor 488	1:300	Thermo Fisher Scientific, A11001
	Goat anti-mouse Alexa Fluor 568	1:300	Thermo Fisher Scientific, A11004
	Donkey anti-goat Alexa Fluor 568	1:300	Thermo Fisher Scientific, A11057

**Table 4 cells-11-00124-t004:** Antibodies used in the organoid immunohistochemistry. Antibodies produced by Abcam (Cambridge, UK), WAKO (Fujifilm Wako Chemicals USA, Richmond, VA, USA), Cell Signaling Technology (Danvers, MA, USA) and Thermo Fisher Scientific.

Antibody	Clone	Host	Pretreatment	Dilution	Cat. No	Producer
IBA1	Polyclonal	Rabbit, IgG	-	1:500	ab153696	Abcam
	Polyclonal	Rabbit, IgG	-	1:500	01919741	WAKO
	GT10312	Mouse, IgG2	Na-citrate	1:500	MA5-27726	Thermo Fisher Scientific
PU.1	Polyclonal	Rabbit	Na-citrate	1:200	2266	Cell Signaling Technology
CD41	M148	Mouse, IgG2a	-	1:1000	ab11024	Abcam
TBR2	WD1928	Rabbit	Triton	1:100	14-4877-82	Thermo Fisher Scientific
DCX	Polyclonal	Rabbit	Triton	1:200	4604	Cell Signaling Technology
SATB2	Polyclonal	Rabbit	Na-citrate	1:500	ab34735	Abcam
TUJ1	TUJ1	Mouse, IgG2a1 k	Triton	1:200	801202	Biolegend
PSD95	D27E11	Rabbit, IgG	Na-citrate	1:200	3450T	Cell Signaling Technology
SYP	SP11	Rabbit, IgG	Na-citrate	1:250	MA5-14532	Thermo Fisher Scientific

## Data Availability

All relevant data is available from the authors upon reasonable request.

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
