# Peer review of "Microglia-like Cells Promote Neuronal Functions in Cerebral Organoids"

_cells, 2021, doi:10.3390/cells11010124_

Round 1

Reviewer 1 Report

This paper is the investigation of EMPs using cerebral organoids. This research is very interesting and valuable for brain biology or tissue engineering researchers. However, some sentences or descriptions are lacking. For example, the enhancement of cell function by using three-dimensional culture has been reported in various tissues. The authors should introduce the concept and discuss this study comparing related recent papers. In addition, one of the most important to use organoids is the drug effect evaluation or in vivo experiment (of course, I understand the assessment of cell function is also essential). The authors should mention these points. Taken together, major revisions should be made. The paper would be re-considered only when all the comments were responded.

  1.  

Introduction and Discussion

There are many reports on the 3D tissue models for cell function, such as migration. Unfortunately, the introduction of this field is too poor. The authors should add some sentences for the description of the fields. To reduce the authors’ burden, I suggest at least these recent papers be added for revision (review and research paper).

Liver

Advanced Drug Delivery Reviews 132 (2018) 296–332

Journal of Physiology and Biochemistry volume 75, pages489–498 (2019)

Cancers

Cancers 202012(10), 2754

Tissue Eng. Part C Methods 201925, 711–720. https://doi.org/10.1089/ten.tec.2019.0189

Brain

Annu. Rev. Biomed. Eng. 2021. 23:359–84

Biomaterials 180 (2018) 117-129

Skin

https://doi.org/10.1016/j.bjps.2017.12.006

Adv. Healthcare Mater. 2017, 6, 1601101

  1. About the organoid size

Hypoxia-derived cell death in organoids should be mentioned.

  1. Overall,

Is this function revealed for the first time in this paper? The points are a little vague.

Author Response

Point 1: There are many reports on the 3D tissue models for cell function, such as migration. Unfortunately, the introduction of this field is too poor. The authors should add some sentences for the description of the fields. To reduce the authors’ burden, I suggest at least these recent papers be added for revision (review and research paper).

Response to point 1:

Thank you for kindly pointing out this gap in the introduction and suggesting relevant references to fill it. Please see the new opening paragraph of the introduction covering the relevance of 3D cell culture in in-vitro research.

Point 2: About the organoid size. Hypoxia-derived cell death in organoids should be mentioned.

Response to point 2: 

Indeed, thank you for pressing this point, it is a central feature of organoids that have been allowed to grow without intervention regarding oxygen and nutrient supply. We have addressed this topic in the discussion on lines 713-717.

On that note, and actually a bit outside the scope of this manuscript, we have performed tunel stainings to visualize the breadth of the necrotic core in organoids, and saw that it is indeed extensive, covering over 40% of the area of the section. Furthermore, the distance from the edge of the organoid to the edge of the necrotic core was not uniform within the section, indicating that the emergence of necrotic tissue is dictated not only by availability of oxygen and nutrients, but perhaps some secondary effects are inducing perinecrotic cell death in asymmetric fashion.

Point 3: Is this function revealed for the first time in this paper? The points are a little vague.

Response to point 3:

Thank you, this is a relevant point to clarify, we added extensive new text in the introduction to address this (lines 82-89). There is only one study to our knowledge covering organoids enriched with microglia (Popova et al., 2021) upon which some functional electrophysiological investigation was performed as a read out. These authors reported just a single comparative developmental point in MEA recordings (sampling the impact of microglial incorporation, just once after 5 weeks following microglial incorporation) after having used a different incorporation technique that we have (enrichment at 15 weeks, unlike our study which is a co-development of the microglial and neuronal population). Our study is unique as it has focused specifically on the emerging, functional, single-cell electrophysiological properties of neuronal entities grown in organoids and the role of microglial enrichment in modulating those functional properties over developmental time in- vitro using whole-cell and MEA electrophysiological read outs. Hence, our study here is unique into shedding light on these neurodevelopmental aspects and the impact of microglia on the organoid maturation over time. To deepen textual clarity we have also added a survey of the functional electrophysiological studies performed in organoids for clarity to the readership in the introduction since they are of interest to the characterization of organoids in-vitro and relevant to our general methodological work.

Reviewer 2 Report

This is a very interesting study in which the authors describe how ti generate a brain organoids contains microglial-like cells. This is an important topic as microglia are essential cells in supporting neuronal maturation and have been found implicated in many brain disorders. The study is well presented and especially the data showing that  microglia-like cells within the organoids promote neuronal and network maturation is very relevant and well-performed. 

Some points that should be addressed:

  1. The reason why certain experiments have only been performed on a subset of IPSC is not clear. This is now indicated in the Material and methods section, it could also be added in the legends of each figure.

  1. Whereas the electrophysiology data are well done and convincing, the immunocytochemistry data (Fig 2) are poorly presented and of low quality, lacking certain controls
  2. Fig 2D. PV staining is not very convincing, adding a nucleaus staining would at least show that the red signals are cells.

  1. How do they explain IBA+ cells in the -ORG group?

  1. Line 419 this should reference to Fig 2a-b not to S Fig 2a-b, same for line 124

  1. Figure 2c, S1 d-e. I’m not entirely convinced whether these are real PSD95 puncta versus “noise”. Was the antibody tested for specificity? What is the meaning on intracellular PSD95 puncta in cell soma and what is the evidence that it is localized at the synapse in these images?

  1. In panel S1e the Y- and X-axis labeling is unclear, what is meant with count and prominence?

Spelling in figure S1e should be prominence instead of prominance

  1. Figure S2 only shows characterization of 1 IPSC line, this should be shown for all lines.

  1. Line 466, data on intrinsic properties are not show. Since the functional characterization is one of the important points of this manuscript, the authors should present these data. This is the only way to asses maturity of the cells. In general “data not shown” is not acceptable anymore.

Author Response

Point 1: The reason why certain experiments have only been performed on a subset of IPSC is not clear. This is now indicated in the Material and methods section, it could also be added in the legends of each figure.

Response to point 1:

We have now included the cell line information in the legends. Thank you for opening this important discussion on using multiple iPCS lines in generating the results. We do see subtle differences in generation of microglia and organoids between individual iPCS lines, as I would expect to see in any differentiation protocol, and it would be an intriguing line of study to identify the factors giving rise to iPCS-line dependent differences.

Still, for the sake of feasibility of the work, we chose to have the key results from multiple lines, while supporting evidence, especially the more tedious cell morphology quantification, from only one line.

Also, coming back to the matter during the revision, we noticed that the Table 1 gave misleading information on the usage of iPCS lines, declaring only the lines that were employed to generate the data shown in the figures, while missing to declare which cell lines were used to replicate the findings. While this impacts only the figure 1, it actually introduces a fourth iPCS line that we used show the establishment of the IBA1+ population.

Point 2: Whereas the electrophysiology data are well done and convincing, the immunocytochemistry data (Fig 2) are poorly presented and of low quality, lacking certain controls

Response to point 2:

Thank you for your critical consideration of IHC results describing interaction between IBA1+ cells and synaptic material. We have now addressed this, together with the review comment #6 in the first paragraph of the section “3.2. IBA1+ cells interact with pre- and post-synaptic elements” (lines: 435-447) and in the new supplement figure 2, which compares the fluorescence signals with or without primary antibody.

Point 3: Fig 2D. PV staining is not very convincing, adding a nucleaus staining would at least show that the red signals are cells.

Response to point 3:

Thank you for prompting us to revisit the PV staining and its role in the manuscript. We decided to withdraw it from the revised version of the manuscript (Fig. 2 d in the original manuscript), thus the Figure 2 of the revised manuscript only contains sub-figures a-c. The PV staining was, to begin with, the least important part of our results. It showed proximity between the IBA1+ and PV+ cells, and thus merely suggested interaction between the two cell types. This might have given the impression that we were trying claim more than we could justify with the data. The practical factor behind the withdrawal is that we had not included the DAPI channel in the IBA1+PV images, and our short period of revision unluckily overlapped with the annual maintenance of the confocal microscopes in our institute. Thus, acknowledging the low information value of the staining, even if intriguing, we deemed it easiest just to drop out this figure. 

Point 4: How do they explain IBA+ cells in the -ORG group?

Response to point 4: Thank you for raising this question, we have addressed this on the lines 390-392. Cerebral organoids are derived from embryoid bodies presenting all the three embryonic layers. Despite the components in the differentiation media favouring the development of the neuroectoderm, it seems that mesodermal cells survive long enough to supply the developing neuronal tissue with some microglia progenitors. Ormel et al 2018 exploited this in their work and changed the protocol slightly to allow more microglia progenitors to migrate to the developing neuronal tissue (by decreasing concentration of heparin 1:10 on days 6-13 in-vitro and postponing the embedding to Matrigel and change to neuroexpansion medium (the same as Diff 1 in our protocol) from d11 to d13)

Point 5: Line 419 this should reference to Fig 2a-b not to S Fig 2a-b, same for line 124

Response to point 5: 

Thank you for pointing out, corrected.

Point 6: Figure 2c, S1 d-e. I’m not entirely convinced whether these are real PSD95 puncta versus “noise”. Was the antibody tested for specificity? What is the meaning on intracellular PSD95 puncta in cell soma and what is the evidence that it is localized at the synapse in these images? 

Response to point 6: 

Thank you for your vigilance with these figures. We would like to refer to our answer for the comment #2. Also, we wish to express our gratitude especially in this question, since it prompted us to acquire deeper understanding on the relevance of PSD95 and conduct more thorough comparison between our IHC signal in the organoids to those published earlier in-vivo.

Point 7: In panel S1e the Y- and X-axis labeling is unclear, what is meant with count and prominence?

Spelling in figure S1e should be prominence instead of prominence

Response to point 7: 

Thank you for pointing this out. We have now elaborated it further in the end of “3.2. IBA1+ cells interact with pre- and post-synaptic elements” and in the legend of Figure S1. The x-axis title corrected to “prominence threshold” to make the meaning clearer to the reader.

Point 8: Figure S2 only shows characterization of 1 IPSC line, this should be shown for all lines.

Response to point 8: 

An iPCS line is typically characterized only in the first publication showing data generated using the line. This manuscript is the first one showing data coming from MAD6. The other three have been published earlier, with their respective citations appearing in the methods section and Table 1.

Point 9: Line 466, data on intrinsic properties are not show. Since the functional characterization is one of the important points of this manuscript, the authors should present these data. This is the only way to asses maturity of the cells. In general “data not shown” is not acceptable anymore.

Response to point 9: 

We thank the reviewer for the comment. We have now replaced the” data not shown” in the text with a new panel in figure 3c as requested

Round 2

Reviewer 1 Report

OK.  Good paper by the appropriate revision.

Reviewer 2 Report

The authors have answered all my questions.